

# Physical and functional properties of fish gelatin-based film incorporated with mangrove extracts

Rahmi Nurdiani[1,2], Rica D.A. Ma'rifah[2], Ihda K. Busyro[2], Abdul A. Jaziri[1,2,3], Asep A. Prihanto[1,2,3], Muhamad Firdaus[1,2], Rosnita A. Talib[4] and Nurul Huda[5]

[1] Department of Fish Product Technology, Faculty of Fisheries and Marine Science, Universitas Brawijaya, Malang, East Java, Indonesia
[2] Bioseafood Research Group, Faculty of Fisheries and Marine Science, Universitas Brawijaya, Malang, East Java, Indonesia
[3] Halal Thoyib Research Centre, Universitas Brawijaya, Malang, East Java, Indonesia
[4] Department of Process and Food Engineering, Faculty of Engineering, Universiti Putra Malaysia, Kuala Lumpur, Malaysia
[5] Faculty of Food Science and Nutrition, Universiti Malaysia Sabah, Kota Kinabalu, Sabah, Malaysia

Corresponding author
Rahmi Nurdiani,
rahmi_nurdiani@ub.ac.id

## ABSTRACT

**Background**. The fishery processing industry produces a remarkable number of by-products daily. Fish skin accounts for one of the significant wastes produced. Fish skin, however, can be subjected to extraction to yield gelatine and used as the primary raw material for edible film production. To increase the functionality of edible films, bioactive compounds can be incorporated into packaging. Mangroves produce potential bioactive compounds that are suitable as additional agents for active packaging. This study aimed to create a fish gelatine-based edible film enriched with mangrove extracts and to observe its mechanical and biological properties.

**Methods**. Two mangrove species (*Bruguiera gymnorhiza* and *Sonneratia alba*) with four extract concentrations (control, 0.05%, 0.15%, 0.25%, and 0.35%) were used to enrich edible films. The elongation, water vapour transmission, thickness, tensile strength, moisture content, antioxidant and antibacterial properties of the resulting packaging were analysed.

**Results**. The results showed that the mangrove species and extract concentration significantly affected ($p < 0.05$) the physical properties of the treated films such as elongation (16.89–19.38%), water vapour transmission (13.31–13.59 g/m$^2$), and active packaging-antioxidant activities (12.36%–60.98%). The thickness, tensile strength, and water content were not significantly affected. Potent antioxidant activity and relatively weak antimicrobial activity of this active gelatine packaging were observed.

# INTRODUCTION

With the increasing eco-friendly awareness of packaging worldwide, biodegradable plastics are the primary focus as replacements for synthetic polymers (*Chin, Lyn & Hanani, 2017*).

In terms of food packaging, natural films are alternative approaches to dealing with serious environmental issues and food safety aspects, representing nontoxic biodegradable films, sustainable materials, and economic resources (*Barbosa-Pereira et al., 2014*; *Adilah, Jamilah & Hanani, 2018*). Furthermore, biopolymer-based films provide a barrier between the outside and inside packaging to maintain food product quality (*Menezes et al., 2019*). Interestingly, biopolymer films can act as carriers of active ingredients such as antimicrobial and antioxidant agents (*Arfat et al., 2014*; *Bahram et al., 2014*; *Lee, Yang & Song, 2016*; *Hanani, Yee & Khaizura, 2019*). These active agents, which are found in the packaging film, can be released onto the food surface and increase the lifespan by decreasing lipid oxidation and microbial growth (*Lee, Yang & Song, 2016*).

Among various kinds of biopolymers, proteins are preferable in developing active packaging due to their greater gas barrier and mechanical properties (*Hanani, Roos & Kerry, 2014*). One of the significant protein-derived films is based on gelatine. This substance is produced by partial hydrolysis of collagen to form films suitable for food packaging (*Menezes et al., 2019*). Traditionally, gelatine is produced from pork skin (41%), cowhide (28.5%), and cow bones (29.5%) (*Hayatudin, 2005*). Poultry skin and feet are also abundant sources for collagen/gelatine production (*Huda et al., 2013*; *Kuan et al., 2017*). However, gelatine sources from certain mammals and poultry may present sensitive religious concerns due to prohibited use in Islam, Judaism, and Hinduism. Hence, growing market demand for alternative gelatine sources, *i.e.,* fish gelatine, has emerged (*Karayannakidis & Zotos, 2016*). Several studies have explored gelatine from different fish species such as catla, rohu, farmed giant catfish, Nile tilapia, rainbow trout, octopus, unicorn leather jacket, pangasius, and starry triggerfish (*Madhamuthanalli & Bangalore, 2014*; *Shyni et al., 2014*; *Jongjareonrak et al., 2010*; *Zeng et al., 2010*; *Tabarestani et al., 2010*; *Jridi et al., 2013*; *Kaewruang, Benjakul & Prodpran, 2014*; *Pradarameswari et al., 2018*; *Muyasyaroh & Jaziri, 2020*). Starry triggerfish possess thick and tough skin, which can be considered a good source of collagen and gelatine (*Jaziri, Muyasyaroh & Firdaus, 2020*).

Although much work has been performed to date, as an important biomaterial of films, fish gelatine can be brittle due to the existence of hydrogen bonds, disulfide bonds, hydrophobic interactions, and electrostatic forces (*Nuanmano, Prodpran & Benjakul, 2015*). Therefore, to enhance the mechanical properties of fish gelatine films, natural compounds can be introduced to produce better packaging films. In addition, the presence of natural compounds can increase the functional properties of films (*Chin, Lyn & Hanani, 2017*). Several authors have reported the use of natural extracts to enrich edible film from fish gelatine. Citrus oil, green tea, *Moringa oliefiera* leaf extract, and aloe vera gel extract were used by *Tongnuanchan, Benjakul & Prodpran (2012)*, *Wu et al. (2013)*, *Lee, Yang & Song (2016)* and *Chin, Lyn & Hanani (2017)*, respectively, and showed significantly enhanced bioactivity of the resulting active packaging. Recently, pomegranate extract and mango kernel extract have also been used for manufacturing active packaging from fish gelatine (*Hanani, Yee & Khaizura, 2019*; *Adilah, Jamilah & Hanani, 2018*). Nevertheless, the use of mangrove extract as an active agent has been limited.

Mangrove plants are known to contain bioactive compounds, which can potentially be incorporated into fish gelatine films. *Sonneratia alba* belongs to the family *Lythraceae*

and has white and pink flowers, fissured bark, dark green fruits, and leaves. This tree can be found in sandy seashores and tidal creeks (*Bojo, 1995*). *Bruguiera gymnorhiza* belongs to the family Rhizophoraceae, which typically has a glabrous, smoothish trunk with reddish-brown bark. This tree can be found on the seaward side of mangrove swamps. As candidates for active compounds incorporated into fish gelatine films, *S. alba* leaves have been shown to exhibit antimicrobial activity against *Escherichia coli*, *Staphylococcus aureus*, and *Bacillus cereus*, with large inhibition zones of approximately 17.5 mm, 12.5 mm, and 12.5 mm, respectively (*Saad et al., 2012*), while *B. gymnorhiza* leaf extract showed an area of inhibition ranging from 8.4 to 15.9 mm against the tested bacteria (*Haq et al., 2011*). Both mangrove plants also had significant radical scavenging activity, with $IC_{50}$ values of 37.23 and 87.5 µg/mL in *B. gymnorhiza* and *S. alba* leaf extracts, respectively (*Nurjanah Jacoeb, Hidayat & Shylina, 2015*; *Gawali & Jadhav, 2011*). Hence, the potency of mangrove extracts as antioxidant and antimicrobial agents in active packaging is quite promising; further investigation is necessary. This study was designed to characterize gelatine films extracted from fish by-product and incorporated with mangrove extracts to develop active packaging.

## MATERIALS & METHODS

### Materials

Starry triggerfish (*Abelistes stellaris*) skins were provided from a small-scale fish industry of the District of Tuban, East Java, Indonesia, and delivered in iceboxes, with a fish/ice ratio of 1:3 (w/w), to the Faculty of Fisheries and Marine Science, Universitas Brawijaya. Upon arrival, the samples were cut into small pieces ($0.5 \times 0.5$ cm$^2$) using clean scissors. The samples were stored in a freezer ($-20\,^\circ$C) until use. Mangrove (*B. gymnorhiza* and *S. alba*) leaves were obtained from the Sendang Biru coastal area, Malang, East Java, Indonesia. 1,1-Diphenyl-2-picrylhydrazyl (DPPH) was purchased from Sigma–Aldrich (Germany). Sodium hydroxide (NaOH), acetic acid (CH$_3$COOH), and methanol (CH$_3$OH) were procured from Merck (Germany). Mueller-Hilton agar was supplied by Sigma–Aldrich (Germany). All the solvents and chemicals used for this study were of analytical grade.

### Fish gelatine extraction

The extraction of gelatine involved an acid-aided process following the method of *Gudmunson & Hafsteinsson (1997)*. First, the frozen skins were thawed with running water. The prepared samples were then soaked in 0.1 M NaOH with a sample-to-solution ratio of 1:5 (w/v). The mixture was stirred for 2 h at $30 \pm 2\,^\circ$C. The alkaline solution was replaced hourly. After being washed with dH$_2$O (pH 7), the samples were soaked in a solution of 0.6 M acetic acid at a 1:5 (w/v) ratio for 2 h. Acid-soaked skin was then washed with dH$_2$O (until pH 7). Afterwards, the samples underwent an extraction process with dH$_2$O (pH 7) at a sample-to-solution ratio of 1:3 (w/v). The extraction process was carried out in a water bath for 4 h at 55 $^\circ$C. The mixtures were filtered with filter paper. The filtrates were then dried in an oven for 48 h at a temperature of 55 $^\circ$C. To obtain gelatine powder, the dried gelatine sheets were ground.

## Preparation of mangrove leaf extracts

The leaves of *B. gymnorhiza* and *S. alba* were authenticated by the botanist from Clungup Mangrove Conservation Centre, Malang Indonesia prior used for experiment. The leaves were then air-dried for three weeks until the moisture content was less than 5% before they were pulverized and weighed for the experiment (*Paputungan, Wonggo & Kaseger, 2017*). The extraction was started by adding methanol (90% v/v) to 100 g of crushed leaves at a ratio of 1:3 (w/v). The samples were extracted using the maceration method for 72 h at room temperature. The solution was replaced daily with fresh methanol. The collected solutions were filtered using Whatman no. 40 filter paper. A vacuum rotary evaporator (set at 40 °C) was used to obtain crude extracts of mangrove leaves. The collected mangrove extracts were then kept at 4 °C in a chiller until further use.

## Preparation of active films

The active film solutions were prepared by dissolving 5 g of fish gelatine with 100 mL of distilled water under magnetic stirring at 50 °C for 30 min. The mixtures were added to 0.5% (v/v) glycerol with heating at 45 °C for 15 min with continuous stirring. To find the best treatment of the active films, different mangrove leaf extract concentrations ranging from 0% to 0.35% (w/v) were added to the film solution. The mixtures were continuously stirred until the solution was thoroughly mixed. The mixtures were then poured onto a $20 \times 20$ cm$^2$ pan and oven-dried using an electric oven at 55 °C for 18–20 h.

## Determination of amino acid composition

The amino acids were analysed using ultra-performance liquid chromatography (UPLC). The hydrolysis of the gelatine sample was carried out in an acidic environment using 6 N HCl heated at 110 °C for 22 h. As an internal standard, $\alpha$-amino butyric acid (AABA) was used. The AccQ-Fluor reagent kit was used as a reagent for amino acid derivatization. The UPLC conditions were as follows: column: ACCQ-Tag Ultra C-18, flow rate: 0.5 mL per min with an injection volume of 1 μL. The elution system was carried out in a gradient with a specific composition (*Waters, 2016*).

## Thickness

The thickness of the film was measured using a screw thread micrometer (Mitutoyo Corp., Japan). Five separate measurements were taken (including the centre spot as the reference point) on the $20 \times 20$ cm$^2$ edible film area. The thickness value of the edible film measured was equal to the average of the five measurements (*Chae & Heo, 1997*).

## Mechanical properties

Tensile strength (TS) and elongation at break (EAB) percentages were measured using an SSB 0500 model Tensile Strength and Elongation from Tester Industries based on a previously reported method (*ASTM, 2003*). Prior to measurement, active films were conditioned in a desiccator with 75% relative humidity (RH) for 24 h. The maximum force value for cutting the measured film was observed on the tool display. The strength was determined based on the maximum load at the time the film broke, while the percentage of elongation was calculated based on the film's length when the film broke. The active

packaging surface area was measured according to a $3 \times 7$ cm$^2$ area. TS can be calculated using the following formula:

$$\text{TS} = \frac{F}{A}$$

where F is the tensile stress (N) and A is the cross-sectional area (mm$^2$).

## Water vapour transmission rate (WVTR)

The WVTR of each film was measured according to (*ASTM, 1987*). The active packaging was cut with a diameter of 3.5 cm and placed between the containers (glass drinks). The first container contained water, while the second container was given silica gel with a constant weight. The samples were set aside for approximately 24 h, and water vapour transmission was calculated according to the following formula:

$$\text{WVTR}\left(\frac{g}{m2day}\right) = \frac{W}{A}$$

where W is the slope of weight gained against time plot (g), and A is the area of film (m$^2$).

## pH

The procedure for testing the degree of acidity (pH) was as follows: 0.2 grams of sample was dispersed in 20 mL of distilled water at a temperature of 80 °C. The sample was then homogenized using a magnetic stirrer. The measurement of the degree of acidity (pH) was carried out using a pH meter when the sample reached room temperature (*BSI, 1975*).

## Moisture content

The moisture content was measured according to the method of Indonesian National Standard (SNI 06-3735-1995). Five grams of active packaging was weighed and placed in an empty cup which were previously dried in the oven and cooled in a desiccator. The cup containing the sample was then covered and put in an oven at 100−102 °C for 6 h.

The moisture content was expressed by the percentage of weight lost during drying according to the equation $(W_0 − W_1/W_0) \times 100\%$, where $W_1$ represents the weight of the film after drying (g) and $W_0$ represents the initial weight of the film (g).

## Determination of antioxidant activity

The DPPH radical-scavenging activity was determined according to the method published by *Hanani, Yee & Khaizura (2019)*. Approximately 25 mg of active packaging was weighed and dissolved in five mL ethanol to obtain a solution. 0.1 mL of the sample solution was then mixed with 3.9 mL of 0.01 mmol L$^{-1}$ DPPH. Each mixture was homogenized and incubated for 30 min in the dark at room temperature. The absorbance value was measured using a UV-Vis spectrophotometer at 517 nm. The percentage of radical scavenging was calculated by dividing the difference of absorbance values of control ($A_{control}$) and the sample ($A_{sample}$) by the absorbance of the control, using the equation $(A_{control} - A_{sample}/A_{control}) \times 100\%$.

## Determination of antibacterial activity

The antibacterial activity was determined by the method of *Hanani, Yee & Khaizura (2019)*. The bacterial strains used in this study consisted of *E. coli, B. subtilis, Salmonella* sp., and *S. aureus*. Prior to inhibition testing, the active film solution was prepared from 5 g of dried fish gelatine with the addition of 0.5% (v/v) glycerol in 100 mL of distilled water. Different concentrations of mangrove extract (0, 0.05, 0.15, 0.25, and 0.35% (w/v)) were added to the mixed solution. Afterwards, blank discs (six mm in diameter) were immersed in the different treated solutions for 15 min. The treated discs were laid on the Petri discs inoculated by four strains in Muller-Hilton agar ($3 \times 10^8$ cfu/mL), and the treatments were then incubated at 37 °C for 16 h. The diameter of the inhibition zones was measured using the following formula:

Zone of inhibition (mm) = zones free of bacterial growth –diameter of the disc.

## Statistical analysis

All experiments were subjected to analysis of variance (ANOVA) using a general linear model, and mean comparisons were applied using a Tukey test. Except for antibacterial experiments, data are presented as the mean from three independent experiments ± SD of the results. Statistical analysis was carried out using Minitab, Version 18, a statistical software program (Minitab Pty Ltd., Sydney, NSW, Australia). The significant level was set at $p < 0.05$.

# RESULTS

## Amino acid composition of fish gelatine

The amino acid composition of *A. stellaris* skin gelatine is tabulated in Table 1. The presence of L-glycine (29.60%) and L-proline (12.53%) was higher than that of other amino acids in *A. stellaris* gelatine.

## Appearance of the films

The appearance of active packaging from *A. stellaris* gelatine film and different concentrations of mangrove extracts is shown in Fig. 1. A higher concentration of leaf extracts resulted in a darker colour of the active packaging.

## Thickness

The thicknesses of fish gelatine films containing two species of mangrove extracts are shown in Table 2. The average thickness of all the films was in the range of 137–143 μm, and the highest film thickness was observed in the gelatine-based film containing 0.35% *B. gymnorhiza* extract.

## Tensile strength (TS)

Table 2 presents the TS values of fish gelatine added mangrove extracts with different concentrations. Based on statistical analysis, the type of mangrove leaves did not significantly affect the TS of active packaging. In contrast, the extract concentration and interaction between mangrove leaf types and extract concentration gave significantly different results ($p < 0.05$).

**Table 1** Amino acids composition (in %) of *A. stellaris* gelatin (skin) and commercial gelatin.

| Amino Acid | *A. stellaris* gelatin (skin) | Commercial gelatin |
|---|---|---|
| L-Serine | 4.42 | 2.18 |
| L-Glutamic Acid | 8.98 | – |
| L-Phenylalanine | 3.03 | 1.92 |
| L-Isoleucine | 1.24 | 1.13 |
| L-Valine | 3.25 | 1.60 |
| L-Alanine | 9.15 | 10.24 |
| L-Arginine | 10.57 | 8.95 |
| L-Glycine | 29.60 | 23.01 |
| L-Lysine | 3.50 | 2.80 |
| L-Aspartic Acid | 4.88 | 4.93 |
| L-Leucine | 2.62 | – |
| L-Tyrosine | 0.83 | 0.15 |
| L-Proline | 12.53 | 12.34 |
| L-Threonine | 4.00 | 2.87 |
| L-Histidine | 1.38 | 0.03 |

**Notes.**

*A. stellaris* skin gelatin was used in this study; commercial gelatin (*Nurilmala, 2004*).

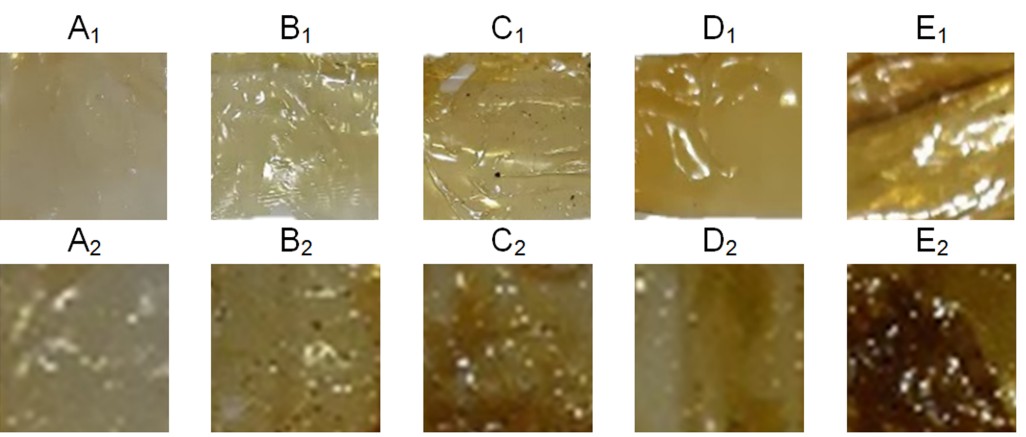

**Figure 1** Appearance of active packaging from A. stellaris gelatin film and different concentration (A: 0%, B: 0.05%; C: 0.15%; D: 0.25%; E: 0.35%) of mangrove extracts. $A_1$-$E_1$: the films incorporated with B. gymnorhiza extract. $A_2$-$E_2$: the films incorporated with S. alba extract.

## Elongation at break (EAB)

Based on the results in Table 2, the type of mangrove leaves and the concentration of mangrove leaf extract significantly affected the EAB values of the active packaging ($p < 0.05$).

## Water vapour transmission rate (WVTR)

The WVTR of packaging films incorporated with 0−0.35% mangrove extracts is shown in Table 2. The results showed that two species of mangroves (*B. gymnorhiza* and *S. alba*) and

**Table 2  Characteristics of active packaging from *A. stellaris* gelatin film incorporated with different concentration of mangroves extracts.**

| Mangrove | Cons. | Thickness (μm) | TS (MPa) | EAB (%) | WVTR (g/m²) | pH | Moisture content (%) |
|---|---|---|---|---|---|---|---|
| *B. gymnorhiza* | 0% | 140.47 ± 0.06[d] | 12.01 ± 0,04[d] | 16.89 ± 0,04[f] | 13.49 ± 0.22[bc] | 6.55 ± 0.07[a] | 6.00 ± 0.24[ab] |
| | 0.05% | 141.26 ± 0.14[e] | 11.94 ± 0,04[e] | 17.04 ± 0,14[de] | 13.40 ± 0.23[bc] | 6.25 ± 0.70[b] | 6.25 ± 0.71[ab] |
| | 0.15% | 141.94 ± 0.04[e] | 11.90 ± 0,04[c] | 17.15 ± 0.12[def] | 13.32 ± 0.20[c] | 6.07 ± 0.14[bc] | 6.33 ± 0.47[ab] |
| | 0.25% | 142.43 ± 0.02[e] | 11.87 ± 0,04[b] | 17.13 ± 0.09[ef] | 13.34 ± 0.11[c] | 6.07 ± 0.21[bc] | 6.75 ± 0.94[ab] |
| | 0.35% | 143.16 ± 0.04[e] | 11.85 ± 0,04[a] | 17.19 ± 0.12[def] | 13.31 ± 0.14[c] | 5.87 ± 0.14[cd] | 6.67 ± 0.71[a] |
| *S. alba* | 0% | 137.63 ± 0.29[g] | 10.39 ± 0.04[g] | 17.51 ± 0.03[d] | 13.59 ± 0.05[c] | 6.60 ± 0.14[ab] | 6.00 ± 0.27[b] |
| | 0.05% | 137.88 ± 0.47[f] | 11.33 ± 0.18[f] | 17.87 ± 0.02[c] | 13.56 ± 0.03[c] | 6.25 ± 0.06[b] | 6.25 ± 0.32[ab] |
| | 0.153% | 138.56 ± 0.06[c] | 12.24 ± 0.01[e] | 18.74 ± 0.02[b] | 13.34 ± 0.16[c] | 6.20 ± 0.14[b] | 6.42 ± 0.17[ab] |
| | 0.25% | 139.68 ± 0.06[c] | 12.78 ± 0.06[e] | 18.95 ± 0.01[b] | 13.32 ± 0.08[a] | 6.05 ± 0.13[cd] | 6.67 ± 0.27[ab] |
| | 0.35% | 140.27 ± 0.01[a] | 13.05 ± 0.02[e] | 19.38 ± 0.03[a] | 13.30 ± 0.04[a] | 5.75 ± 0.17[d] | 6.75 ± 0.17[a] |

**Notes.**

Means in the same column with different superscripts are significantly different ($p < 0.05$).

different concentrations of mangrove leaf extracts were significantly different compared to the WVTR values of the active packaging films ($p < 0.05$).

## pH value

The average pH value of the active packaging ranged from 5.75 to 6.6. The concentration of mangrove leaf extracts significantly affected ($p < 0.05$) the pH. In contrast, there was no significant difference between the mangrove species.

## Moisture content

Table 2 shows that the moisture content of the active packaging ranged from 6−6.50%. Incorporating a higher concentration of mangrove leaf extracts significantly increased the moisture content ($p < 0.05$).

## Antioxidant activity

The antioxidant activity of the edible film in this study was measured using the DPPH radical-scavenging assay. Figure 2 shows the antioxidant activity of films incorporated with *B. gymnorhiza* and *S. alba* extracts. With the increasing concentration of mangroves extracts, the scavenging effect increased significantly ($p < 0.05$) from 12.39% to 57.32% and from 12.36% to 60.98% for *B. gymnorhiza* and *S. alba* extracts, respectively.

## Antibacterial activity

Four bacterial strains were used to determine the antibacterial activities of fish gelatine-derived films incorporated with *B. gymnorhiza* and *S. alba* leaf extracts. The inhibition zones of the films are shown in Table 3. The films with higher concentration of mangroves extracts, showed significantly ($p < 0.05$) increased antibacterial activities.

## DISCUSSION

The gelatine film properties are determined by the source of gelatine regarding the molecular weight distribution and amino acid composition (*Ramos et al., 2016*). Compared

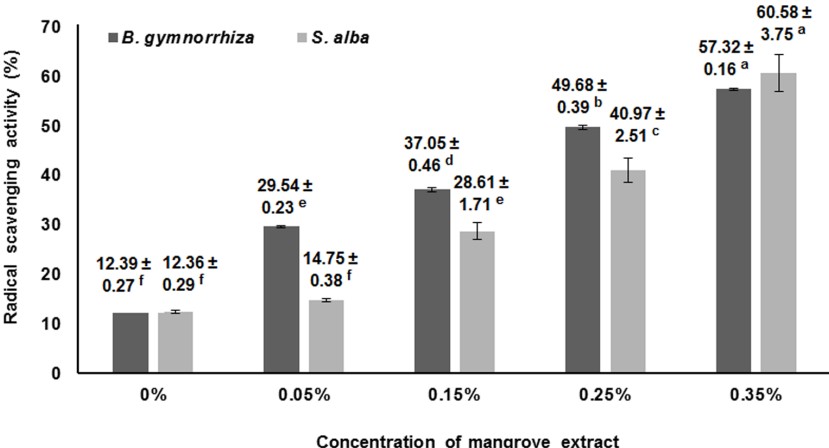

**Figure 2** Antioxidant activity of the fish gelatin-based films incorporated with 0–0.35% *B. gymnorhiza* and *S. alba* leaves extract.

**Table 3** Inhibition zone of the fish gelatin films with 0–0.35% *B. gymnorhiza* and *S. alba* leaves extract.

| Mangrove | Concentration | Diameter of inhibition zone (mm) | | | |
|---|---|---|---|---|---|
| | | *S. aureus* | *E. coli* | *B. subtillis* | *Salmonella* sp. |
| | 0% | – | – | – | – |
| | 0.05% | – | $0.84 \pm 0.05^e$ | – | – |
| *B. gymnorhiza* | 0.15% | $1.43 \pm 0.06^c$ | $1.25 \pm 0.06^d$ | $1.31 \pm 0.06^d$ | $0.74 \pm 0.02^b$ |
| | 0.25% | $1.87 \pm 0.02^b$ | $2.06 \pm 0.08^b$ | $1.84 \pm 0.05^c$ | $1.35 \pm 0.08^c$ |
| | 0.35% | $2.32 \pm 0.08^a$ | $1.96 \pm 0.02^b$ | $2.27 \pm 0.08^a$ | $1.89 \pm 0.06^a$ |
| | 0% | – | – | – | – |
| | 0.05% | $0.42 \pm 0.02^e$ | $0.72 \pm 0.01^e$ | – | – |
| *S. alba* | 0.153% | $1.25 \pm 0.02^d$ | $1.19 \pm 0.02^d$ | $1.44 \pm 0.04^e$ | $0.53 \pm 0.02^b$ |
| | 0.25% | $1.52 \pm 0.03^c$ | $1.80 \pm 0.06^c$ | $1.92 \pm 0.02^c$ | $1.29 \pm 0.03^c$ |
| | 0.35% | $1.99 \pm 0.04^b$ | $2.22 \pm 0.06^a$ | $2.42 \pm 0.03^b$ | $1.76 \pm 0.02^a$ |

**Notes.**
Means in the same column with different superscripts are significantly different ($p < 0.05$).

to commercial gelatine, *A. stellaris* gelatine showed slightly higher concentrations of L-glycine and L-proline. A similar amount of glycine was observed in Sin croaker gelatine (*Cheow et al., 2007*). According to *Benjakul, Kittiphattanabawon & Regenstein (2012)*, glycine represents nearly one-third of the total residues, except for 14 amino acids at the N-terminus and 10 amino acids at the C-terminus. In addition, L-proline is the second most abundant amino acid characterizing gelatine, representing 12% of the total amino acid residues. *Aewsiri, Benjakul & Visessanguan (2009)* and *Hoque, Benjakul & Prodpran (2010)* stated that a lower hydrophobic amino acid (proline and leucine) content in gelatine might result in poorer surface activity.

When *A. stellaris* skin gelatine was incorporated with *B. gymnorhiza* and *S. alba* leaf extracts, it resulted in yellow to dark brown active packaging (Fig. 1). A higher concentration

of mangrove extracts produced a darker film. *S. alba* and *B. gymnorhiza* leaf extracts contain a certain amount of flavonoids, steroids/triterpenoids, saponins, and tannins/phenols known to have physical and functional properties (*Nurjanah Jacoeb, Hidayat & Shylina, 2015*; *Gawali & Jadhav, 2011*). Since *S. alba* extracts had a darker colour than *B. gymnorhiza*, darker films were produced.

The film's mechanical properties are generally affected by the film's physical characteristics, such as thickness. In our study, the mangrove leaf extract concentration significantly affected the thickness of active packaging ($p < 0.05$). It can be seen that the higher the concentration of mangrove leaf extract added, the greater the film thickness. This result was due to the high concentration of the constituent components of active packaging, which can increase the amount of total solids in active packaging to increase the thickness. *Supeni (2012)* stated that as more total solids are added to an edible film, the resulting edible film thickness increases. The gelatine-based film that we produced was thicker than fish gelatine films containing pomegranate peel powder (*Hanani, Yee & Khaizura, 2019*) and aloe vera (*Chin, Lyn & Hanani, 2017*). *Jacoeb, Nugraha & Utari (2014)* stated that thicker active packaging would affect the properties of active packaging. It is more rigid so that the packaged product will be protected from the surrounding environment.

Our study showed that the higher the *S. alba* leaf extract concentration added, the stronger the TS of active packaging. This result might be related to the phenolic compounds found in mangroves (*Banerjee et al., 2008*; *Asha, Mathew & Lakshmanan, 2012*). Polyphenolic compounds contain many hydrophobic groups, which may form hydrophobic interactions with the hydrophobic region of gelatine. In addition, hydroxyl groups of phenolic compounds can combine with the hydrogen acceptors of gelatine molecules by hydrogen bonds (*Gómez-Guillén et al., 2009*). The interaction between protein and polyphenol was reported to increase the mechanical properties of gelatine-based films (*Hoque, Benjakul & Prodpran, 2011*).

The EAB value of fish gelatine-based active packaging (Table 2) ranged from 16.89–19.38%. The more mangrove leaf extract that was added, the greater the EAB value. An increase in TS and EAB values of jellyfish protein films was also reported by *Lee et al. (2015)*, when increasing the wasabi extract concentration added. Similarly, the EAB of pufferfish skin gelatine film increased as a higher concentration of *Moringa oleifera* was added (*Lee, Yang & Song, 2016*). *Warkoyo & Zuhriansyah (2014)* stated that adding more active ingredients could stretch the intermolecular space of the film matrix network and decrease the number of internal hydrogen bonds, thereby reducing the film's fragility and increasing the percent elongation.

The findings showed that the film WVTR values decreased with increasing incorporation of mangrove leaf extracts into fish gelatine films. Similarly, *Adilah, Jamilah & Hanani (2018)* reported that the WVTR values of active packaging films decreased with increasing mango kernel extracts. *McHugh & Krochta (1994)* revealed that the WVTR value of a material is influenced by the presence of chemical material and the structure of the forming material, the concentration of plasticizers, and environmental conditions such as humidity and temperature. Moreover, *Silverajah et al. (2012)* stated that the presence of plasticizers could theoretically reduce the intermolecular strength along the polymer chain, increase

film flexibility, and reduce the barrier properties of films. In addition, air bubbles in the layers and the increase in hydrophilic components can increase the water vapour transmission rate.

The low pH value of active packaging is due to the gelatine used. This result is presumably because residual acid remained after the gelatine extraction process due to an inability of the washing process to remove all the acetic acid used. Thus, this situation affects the final product of active packaging, which has an acidic pH. This result was confirmed by *Trilaksani & Nurilmala M. Setiawati (2012)*, who explained that the pH value was highly dependent on the washing process after acid immersion. Therefore, a complete washing process will reduce the acid content trapped in the skin so that the pH value will be neutral.

The addition of mangrove leaf extracts significantly increased the moisture content of the active packaging. The water content of active packaging is influenced by the total water molecules present in the composite film network microstructure. The study conducted by *Jin-Hua et al. (2014)* reported a higher moisture content in active packaging with the addition of grape seed extract (14.40–24.43%). An increased water content occurs due to the reduction in gelatine-gelatine interactions when active substances are added, consequently increasing the availability of free hydroxyl groups to absorb more water (*Chin, Lyn & Hanani, 2017*).

Without the addition of mangrove extract, the starry triggerfish skin gelatine-based film showed low antioxidant activity (Fig. 2). *Gómez-Guillén et al. (2007)* also reported that tuna skin gelatine inhibited free radicals, as determined by FRAP and ABTS assays. Antioxidant compounds influence the antioxidant activity of active packaging films and the ability of these compounds to reduce free radicals (*Hanani, Yee & Khaizura, 2019*). The higher the concentration of *B. gymnorhiza* and *S. alba* leaf extracts was, the higher the antioxidant activity value obtained. *Chin, Lyn & Hanani (2017)* showed the results of antioxidant activity in active packaging with the addition of *Aloe vera*, with quite large values, between 65.78% and 74.76%. These values are relatively higher than those obtained in this study.

Table 3 shows that the control film caused no inhibition zone of inhibition, while the films with higher concentrations of mangrove extracts (above 0.05%) presented significantly ($p < 0.05$) increased antibacterial activities. In particular, the highest antibacterial activity was recorded against *B. subtilis*, with 2.42 mm inhibition. Fish gelatine films containing fruit pomace extracts showed higher antibacterial activities compared to our study (*Staroszczyk et al., 2020*). According to *Oroh et al. (2015)*, the diameter of the inhibitory zone formed can indicate the antibacterial strength of the extract used in active packaging. The categorization of antibacterial agents with inhibition zones was defined as follows: diameters >20 mm was designated very strong; diameters ranging from 10–20 mm were designated strong; diameters ranging from 5–10 mm were designated moderate; and diameters <5 mm were designated weak. Our results also showed that the antibacterial activity of mangrove extracts was stronger against gram-positive bacteria (*S. aureus* and *B. subtilis*) than against gram-negative bacteria (*E. coli* and *Salmonella*). Similarly, *Hanani, Yee & Khaizura (2019)* reported that pomegranate extracts showed more potent antibacterial activity against gram-positive bacteria.

## CONCLUSIONS

Active packaging films prepared from starry triggerfish skin gelatine incorporated with leaf extracts of *B. gymnorhiza* and *S. alba* were developed. The addition of leaf extract contributed significantly to the physical and mechanical properties of the active packaging. Antioxidant activity of gelatin films increases in a concentration dependent manner with the *B. gymnorhiza* and *S. alba* extracts. High antioxidant property of film would be beneficial in inhibiting oxidation and extending the shelf life of food product. Nevertheless, our study also showed that adding up to 0.35% mangrove extracts resulted in relatively low antibacterial activities. Therefore, further study on the optimum concentration of extracts to improve the functional properties of active packaging is essential.

### Funding

This work was supported by Universitas Brawijaya and the Ministry of Education, Culture, Research and Technology, Indonesia, under the World Class Research grant with contract no: 438.2/UN0/TU/2021. The funders had no role in study design, data collection and analysis, decision to publish, or preparation of the manuscript.

### Grant Disclosures

The following grant information was disclosed by the authors:
Universitas Brawijaya and the Ministry of Education, Culture, Research and Technology, Indonesia, under the World Class Research grant: 438.2/UN0/TU/2021.

### Competing Interests

The authors declare there are no competing interests.

### Author Contributions

- Rahmi Nurdiani conceived and designed the experiments, performed the experiments, authored or reviewed drafts of the paper, and approved the final draft.
- Rica D.A. Ma'rifah and Ihda K. Busyro performed the experiments, analyzed the data, prepared figures and/or tables, and approved the final draft.
- Abdul A. Jaziri conceived and designed the experiments, performed the experiments, prepared figures and/or tables, and approved the final draft.
- Asep A. Prihanto and Muhamad Firdaus analyzed the data, authored or reviewed drafts of the paper, and approved the final draft.
- Rosnita A. Talib conceived and designed the experiments, analyzed the data, authored or reviewed drafts of the paper, and approved the final draft.
- Nurul Huda conceived and designed the experiments, authored or reviewed drafts of the paper, and approved the final draft.

## Data Availability

The raw data is available at Figshare: Nurdiani, Rahmi; Prihanto, Asep Awaludin; Firdaus, Muhammad; Talib, Rosnita A.; Huda, Nurul; Jaziri, Abdul Aziz; et al. (2021): Raw Data Active Packaging R Nurdiani.xlsx. figshare. Dataset. https://doi.org/10.6084/m9.figshare.16607792.v2.

The statistical data is available at Figshare: Nurdiani, Rahmi; Prihanto, Asep Awaludin; Firdaus, Muhammad; Talib, Rosnita A; Huda, Nurul; Ma'rifah, Rica; et al. (2021): Minitab report Active Packaging - Rahmi Nurdiani. figshare. Dataset. https://doi.org/10.6084/m9.figshare.16635865.v1.

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
