# Peer review of "Physical and functional properties of fish gelatin-based film incorporated with mangrove extracts"

_PeerJ, doi:10.7717/peerj.13062_

## Round 0.1 · original submission · Major Revisions

Reviewers have now commented on your paper. You will see that they are advising that you revise your manuscript. If you are prepared to undertake the work required, I would be pleased to reconsider my decision.

Reviewer 1 ·

Basic reporting

no comment

Experimental design

no comment

Validity of the findings

no comment

Additional comments

The manuscript by Rahmi Nurdiani et al, deals Characterization of fish gelatin-based film incorporated with mangrove extracts and determination of its Physical and functional properties. In my opinion, the paper is well written and contributes to the existing knowledge. The following points may be considered while revising the article:

1. The manuscript needs to be edited for its English.
2. Abstract should be included numerical results.
3. What is the best reported for antibacterial activities of fish gelatin-based film incorporated with mangrove extracts by other methods and what are the main cause for introducing mangrove extracts as an effective incorporated materials? For that purpose, the authors can make new table for comparison with other studies with discussion in the main text.

Reviewer 2 ·

Basic reporting

In the manuscript “Physical and functional properties of fish gelatinbased film incorporated with mangrove extracts", the authors aimed to create a fish gelatin-based edible film enriched with mangrove extracts and observe its mechanical and biological properties. The work has been carried out carefully, using the relevant experimental method. However, there are still some concerns needing to be addressed before I can recommend publication of this paper in PeerJ:

1. Some type errors should be carefully checked and avoided in the whole manuscript and English writing should be improved for better understand.
2. All references should be prepared according to publication manual of the PeerJ.

Experimental design

3. In experimental section, methods should be described with sufficient information.

Validity of the findings

4. In discussion section, it would be necessary to compare this result with other fish gelatin-based edible film published previously.

Additional comments

5. Why B. gymnorhiza and S. alba were selected as candidates of active compounds inserted into fish gelatin films? It should be discussed in the Introduction.
6. In my opinion, the characterization of materials was not presented in the proper way, the thermal stability of gelatin-based edible film should be added.

Annotated reviews are not available for download in order to protect the identity of reviewers who chose to remain anonymous.

Reviewer 3 ·

Basic reporting

The manuscript is straight-forward easy to understand albeit with minor typographical errors. Both the results and discussion sections requires major improvement.
Figure 1 is not clear. It should be labelled. Please add a scale/magnification factor.

Experimental design

Standard methodologies were used but some is rather confusing and require further explanation.
The leaves of many plants have been shown to have antioxidant and antimicrobial activities - what is the reason for choosing two mangrove species in this study?
Lines 137-146: name the botanist who authenticated the leaves
Line 153: please explain how the range of extracts' concentration was chosen - this is crucial as in the subsequent parts, the authors stated that low bioactivities were due to "low concentrations of extracts"
Line 199: did the authors measured the pH at 80 degrees celcius?
Lines 214-223: positive control is missing
Lines 225-236: positive control is missing (positive controls are necessary to allow comparison of the results of test substances and some form of validation of the procedure in the assays)
SD/SE is missing in Table 1 - how many times the analysis was performed?
Error bars are missing in Figure 6.

Validity of the findings

The authors presented their results in tables and figures but description of their main findings in each section / table / figure is missing, for instance, there are no elaboration on the results of antioxidant and antimicrobial assays (lines 296-307).
The claims about phenolic compounds in mangrove extracts and hydrophobic polyphenolics (lines 351-362) are not substantiated.
Direct comparison of values obtained in this study vs. those in the literature is not possible due to differences in methodologies (lines 406-408).
Please justify why comparison was made to a study that used a completely different sample than what was used in this study (lines 422-423).
I Figure 6, film without the addition of mangrove extracts showed considerable radical scavenging (10%). Is this possible? Why are there two values reported when there is only one sample of film without any extracts. Please justify.
Lines 429-430: Please rephrase this sentence. "higher antioxidant activities were recorded as a higher concentration of leaf extracts was added to the film" - one need not perform the experiment to deduce this
Lines 430-431: "Our study showed that adding 0.05-0.35% of mangrove extracts resulted in relatively low antibacterial activities" The concentrations were decided by the authors. Again, there is very little comparison of the results obtained in this study with those in the literature.
The authors should revise the conclusion section. A more reasonable conclusion based on the findings in this study should be provided instead.

Additional comments

The manuscript describes the preparation of fish gelatin-based film with addition of mangrove extracts and studies on the functional, antioxidant and antibacterial properties of the resulting film. Findings are interesting but data presentation, analysis and interpretation is weak as explained above. Results need to be compared with the literature in order to make an accurate assessment of how the film fare compared to those that have been reported previously.

---

## Round 0.2 · accepted · Accept

I have completed my evaluation of your manuscript and it gives me great pleasure to inform you that your manuscript is now accepted for publication. Congratulations!